# Investigating the Reliability of Shore Hardness in the Design of Procedural Task Trainers

**DOI:** 10.3390/bioengineering12010041

**Published:** 2025-01-07

**Authors:** Kyleigh Kriener, Kate Sinclair, Grant Robison, Raushan Lala, Hayley Finley, William Jase Richardson, Mark J. Midwinter

**Affiliations:** 1School of Biomedical Sciences, Faculty of Medicine, The University of Queensland, Brisbane, QLD 4072, Australia; kate.sinclair@uq.edu.au (K.S.); w.richardson@student.uq.edu.au (W.J.R.);; 2Ochsner Clinical School, 1401 Jefferson Hwy, Jefferson, LA 70121, USA

**Keywords:** biomechanics, human tissues, physician training, reliability, Shore hardness, simulation design, procedural task trainer

## Abstract

The haptic fidelity of biomimetic materials used in the design of procedural task trainers is of growing interest to the medical community. Shore hardness has been proposed as a method for assessing tissue biomechanics and replicating the results as a way to increase the fidelity of biomimetics to tissues. However, there is limited research on the reliability of human tissue measurements using Shore scales. Using human tissues (internal carotid artery [ICA], internal jugular vein [IJV], vagus nerve [VN], sternocleidomastoid muscle [SCM], and overlying skin [skin]), this study evaluates (1) the inter-rater reliability of Shore hardness measurements, (2) examines the relationship between tissue thickness and hardness, and (3) investigates the impact of a measurement method (freehand vs. durometer stand). Preserved tissues, specifically a liver and components of the anterior triangle of the neck, were extracted from cadavers and measured by three independent raters using digital Shore durometers. Testing revealed that although Shore A demonstrated better inter-rater reliability compared to Shore OO, both scales exhibited poor-to-moderate reliability. ICC values for Shore A ranged from 0.21 to 0.80 and were statistically significant (*p* < 0.05) for all tissue types except the SCM. In contrast, Shore OO demonstrated poorer reliability, with ICC values ranging from 0.00 to 0.41. The ICC values were only significant for the ICA, IJV, and VN (*p* < 0.05). An inverse correlation between tissue thickness and hardness on the Shore A scale was found for all tissues and was significant (*p* < 0.05) for ICA, VN, and skin. There were mixed results for the correlation between tissue thickness and hardness on the Shore OO scale (−0.06–0.92), and only IJV had a statistically significant correlation (*p* < 0.05). Finally, the median hardness values on the Shore OO scale were significantly greater when measured using a durometer stand vs. freehand (Z = 4.78, *p* < 0.05). In summary, when using appropriate standards and addressing the challenges of tissue thickness and variability in freehand measures, Shore hardness has the potential to be used by clinicians in the clinical setting and in the selection of biomimetic materials in the design of task trainers.

## 1. Introduction

There is a growing interest within the medical community to develop biomimetic materials for the creation of task trainers (referred to as trainers or simulations). Over the past two decades, the utilisation of such trainers has gained significant traction, particularly in the context of teaching and assessing medical students and training physicians. The primary educational objective of these trainers is to provide an environment wherein trainees can acquire and refine their skills with reduced risk to patients.

However, a critical concern arises with the fidelity of the tactile feedback (also referred to as haptic fidelity) provided by these trainers [1]. Trainers failing to replicate the tactile feedback encountered in real-world scenarios may result in trainees inadvertently employing aberrant forces or manoeuvres during procedures, potentially leading to patient harm [2]. Such deviations from the proper technique may prolong the learning process because trainees must “unlearn” these erroneous behaviours [2].

To address the challenge of haptic fidelity in simulations, it is essential that the incorporated synthetic materials (also referred to as biomimetics) mimic the biomechanical and physical properties of human tissues [2,3]. However, selecting the appropriate properties for biomimetics presents a complex challenge. The replication of a single tissue type entails a broad spectrum of biomechanical and physical characteristics that can be influenced by an individual’s anatomy, physiology, and body composition. Moreover, the task is further complicated by the scarcity of research elucidating which biomechanical properties specifically influence the perceived tactile feedback during medical procedures.

With the increasing prevalence and affordability of three-dimensional (3D) printing, the mechanical properties of materials utilised in the design of trainers have garnered significant attention [4]. A proposed starting point for assessing the biomechanical properties of human tissue is Shore hardness [4]. By leveraging Shore hardness as a metric, 3D materials can be selected based on comparative Shore hardness readings from the relevant mimicked tissues. Incorporating 3D-printed mimics into trainers may facilitate the creation of lifelike simulations, offering trainees a more representative learning experience.

The general method of measuring Shore hardness employs a durometer, a device that gauges hardness when its metal indenter is pressed onto a material’s surface [5]. The shape of the metal indenter varies based on the (1) type of hardness test (e.g., Vickers, Knoop, Shore, etc.) and (2) the hardness scale (e.g., Shore A, Shore OO, etc.). Hardness measurement quantifies a material’s resistance to permanent (plastic) deformation and has been used to establish empirical relationships between hardness and other mechanical properties such as yield stress and elastic modulus [5].

Aside from offering mechanical insights into human tissues, Shore hardness stands out as an appealing methodology because of its simplicity, speed, and non-destructive nature. In industrial contexts, Shore hardness measurements adhere to standards such as ASTM D2240 [6], which ensure consistent and dependable material data acquisition. However, comparable standards for human tissue testing are notably absent. This absence is likely due to the intricate nature of biological tissues [7]. For instance, when using Shore durometers, ASTM D2240 recommends a minimum material thickness of 6 mm for elastomers [6], a requirement often impractical for tissues like fascia. Moreover, industry standards stipulate minimum distances between both repeated Shore measurements and the material’s edge [6], which are challenging in certain anatomical regions.

Despite its ease of use, there is a paucity of published studies utilising Shore hardness as a methodology. Furthermore, the data from studies employing Shore hardness may be limited in their utility for selecting comparable 3D materials. Table 1 presents data extracted from five distinct studies where Shore hardness was utilised. Notably, the table exhibits limitations in the type of tissues tested and shows wide ranges of the reported values. For instance, concerning normal pancreatic tissues, Tejo-Otero reported an average Shore O value of 13, whereas Belyaev documented median values of 27 and 30 [8,9].

The discrepancy in pancreatic readings between the studies published by Tejo-Otero and Belyaev may stem from adherence (or lack thereof) to the ASTM D2240 standard. Belyaev et al. conducted freehand measurements, a method that may have influenced their readings. The use of a stage, however, is not explicitly detailed in the published methodology by Tejo-Otero [8,9]. The ASTM D2240 standard provides precise guidelines for durometer usage, particularly emphasising the use of durometer stands during measurements. This emphasis is likely due to the potential introduction of factors creating variability in Shore readings when measured freely by hand. Such an example includes inconsistency in the applied force when indenting the test specimen.

As the use of physical simulations grows, particularly in the context of 3D printing and do-it-yourself approaches, the demand for standardised tissue values becomes increasingly important. Whilst Shore hardness is an appealing mechanical metric for material selection in clinical training tools, some studies have briefly explored its ability to inform clinical decision making [9,11,13,14]. Therefore, it is also imperative to establish whether measuring Shore hardness is a reliable method for use among a clinician population, as there are significant research gaps in this area.

To address these research gaps, this study aims to evaluate the reliability of Shore hardness for human tissue measurement by (1) assessing inter-rater reliability across various tissue samples, (2) examining the influence of tissue thickness on Shore hardness readings, and (3) analysing the variability introduced by freehand measurements. It is hypothesised that stage-assisted Shore hardness measurements will exhibit consistency across scales and raters, whereas variations in tissue thickness and freehand techniques are expected to result in greater variability in mean hardness readings.

## 2. Materials and Methods

This section describes the general procedures and statistical methods used to meet the objectives of the study, tissue sample extraction, and characteristics of the study population and raters. Ethical approval for this study was obtained from the University of Queensland (2021/HE002373), and the study was conducted in accordance with the Declaration of Helsinki. This study was registered in the Open Science Framework on 7 May 2023 (https://osf.io/ucxmy). All of the reported information follows the guidelines for reliability and agreement studies by Kottner et al. [15].

### 2.1. Tissue Samples and Study Population

Tissues used to study inter-rater reliability and correlation between tissue thickness and Shore readings were excised from the carotid sheath. Bilateral tissue samples were obtained from formalin-preserved head and neck cadavers after dissecting the anterior triangle of the neck. The characteristics of the cadavers used are summarised in Table 2.

The skin was reflected anteriorly following an oblique incision along the anterior border of the sternocleidomastoid muscle (SCM). A rectangular skin sample of approximately 5 cm long by 2 cm wide was taken from this flap. The superficial cervical fascia was dissected off the surface of SCM, and a plane deep into the muscle was developed. An SCM sample, 5 cm in length, was removed for Shore hardness testing. Blunt dissection of the carotid sheath was then undertaken to visualise the internal jugular vein (IJV), internal carotid artery (ICA), and vagus nerve (VN). These structures were excised as a tissue block of 5 cm in length. The ICA, IJV, and VN were then separated after the removal of the tissue block. The VN was isolated from the tissue block and tested. A longitudinal incision along the ICA and IJV was made so that the structures could be opened and placed flat on the test platform (Figure 1). Clotted blood from the luminal surface of the vessels was removed.

For freehand and stage measurements, a whole liver specimen was excised from a soft embalmed cadaver, free of liver disease. The liver was selected because the surface is relatively homogeneous and samples could be shaped into the desired length, width, and thickness recommended by the measurement standards (i.e., 12 mm × 12 mm × 6 mm) [6].

### 2.2. Shore Measurements and Rater Characteristics

Shore hardness measurements were performed by 3 different raters (GR, KK, and KS) using both Shore OO (±4) and Shore A (±1) scales. Digital durometers (DSD Shore Durometer, Starr, Victoria, Australia) were mounted on a stage containing a 1 kg weight (Figure 2). Prior to obtaining hardness measurements from tissues, the durometer was calibrated using the standardised calibration blocks provided by the manufacturer. The durometer was actuated downwards using a lever, and the foot indented the tissue samples. After a three-second pause, the hardness reading was recorded.

For all tissue samples, each rater performed five different measurements that were evenly spaced along the tissue specimen. The spacing of measurements was distributed evenly by measuring near each corner and in the center of the specimen. Where possible, measurements were taken approximately 12 mm from the edge.

With the exception of the VN, tissue length, width, and thickness were measured using a manual vernier caliper (±0.02 mm). Since the VN was too thin to be longitudinally incised, only length and diameter dimensions were collected. Measurements in each dimension (i.e., length, width, thickness, and diameter) were repeated three times and averaged. Freehand Shore measurements were taken using the Shore OO scale by a rater randomly selected from the 3 raters. Shore A was not used because there are no comparable literature values using a freehand method. The rater received no special instructions on how to obtain freehand measurements. The same rater also collected Shore measurements using the durometer on the stage and was not informed about tissue origin (i.e., the same liver tissue was used for both methods).

Three raters (GR, KK, and KS) were randomly selected from a rater pool consisting of postgraduate students enrolled in the Faculty of Medicine at the University of Queensland (UQ). As suggested in the guidelines by Kottner et al., the rater characteristics are reported in Table 3 [15].

### 2.3. Power Calculation

Using previously collected pilot data from 2022, expected confidence intervals (CIs) and corresponding sample sizes were calculated for absolute agreement ($\hat{p}$) using previous work detailing how to estimate ICC sample size from exact CI widths [16]. The relationship between CI width and the number of samples required to achieve that width given the absolute agreement value for each tissue type is shown in Figure 3. The curves in Figure 3 were calculated based on type I (α) and II (β) errors equal to 0.05 and 0.20, respectively.

In Figure 3, there is logarithmic decay in the CI width as the number of samples increases. The CI widths sharply decline for sample sizes between 0 and 20 and become relatively stable for smaller sample sizes for larger $\hat{p}$ values. Since the VN has the smallest $\hat{p}$ value (0.53), the worst expected CI width is estimated to be between approximately 0.20 (*n* = 200 cadavers) and 1.00 (*n* = 10 cadavers). Institutional projections using past body donations at UQ found that achieving a sample size of 200 cadavers would take a minimum of 3 years to complete this study, requiring more resources than available.

Previous work has defined a “reasonably” narrow CI width ($CI_{Narrow}$), which is shown in Equation (Equation 1). Table 1 uses the measured $\hat{p}$ values in Equation (Equation 1) to estimate a “reasonably” narrow CI width ($CI_{Narrow}$). The $CI_{Narrow}$ was found on the ordinate of Figure 3 and traced across the figure until it intersected with the curve of the specified tissue. At the intersection between the $CI_{Narrow}$ and the tissue curve, the value on the abscissa is reported as the sample size (*n*) in Table 4.
(1)$$CI_{Narrow}=0.40\hat{p}$$

As seen in Table 4, achieving $CI_{Narrow}$ for the two lowest $\hat{p}$ values results in cadaver sample sizes between 100 and 200 cadavers. Such a large number of cadavers is unrealistic due to resource availability. Therefore, a balance was struck between resources and data reliability, which resulted in the selection of a value for *n* that provided “reasonable” reliability for most of the tissue samples (*n* = 14 cadavers). This sample size estimates the CI widths of ICC values to be between 0.15 and 0.80. Whilst the level of precision is not ideal, it was determined that it is sufficient for the study’s primary objective. Implications of the selected sample size are presented in the Discussion section.

### 2.4. Statistical Analysis

Prior to conducting any statistical calculations, the data from each rater, Shore scale (A, OO), and tissue type underwent normality assessment. The Shapiro–Wilk test was utilised for ICC normality testing because the number of samples in each dataset was less than 50 [17]. Homoscedasticity was assessed using Levene’s test prior to ICC calculation. All significance tests in this study were performed at a confidence level of 95%, and all data analysis was performed using MATLAB (R20223b Update 5 (23.2.0.2459199), The MathWorks, Natick, MA, USA).

The primary objective of this study is to ascertain whether Shore hardness is an accurate (or reliable) measure of tissue hardness. Reliability was evaluated using an inter-rater approach, and the intraclass correlation coefficient (ICC) was calculated [15]. This analysis assumes random effects from both the subjects (cadavers from which the tissues were extracted) and the raters (Model 2A, as described by Shrout and Fleiss [18]). In accordance with the ASTM standard [6], absolute inter-rater agreement of the average Shore readings for each tissue type was assessed (classified as (A,k) by McGraw and Wong [19]).

The secondary objective of this study is to explore the relationship between Shore hardness readings and tissue thickness. Given that the data are continuous with underlying normal distributions [20], the Pearson product–moment coefficient of correlation was calculated for each tissue type from each Shore scale.

The final objective is to investigate whether Shore hardness measurements change when measurements are obtained by freehand or using a durometer stand measurements. Given that the freehand measures were not normally distributed and dependent, the paired Wilcoxon signed-rank test was performed. A liver was selected so that the measurements taken in this work could be compared to those in previous studies [8,11].

## 3. Results

### 3.1. Inter-Rater Reliability

A total of 1152 Shore A and Shore OO readings were measured by three different raters, and five different tissue types were tested from a total of fourteen different cadavers. The mean hardness values from all three raters on the Shore A scale were 62.25 (ICA), 75.39 (IJV), 70.23 (VN), 44.12 (skin), and 23.90 (SCM). The mean hardness values on the Shore OO scale were 55.71 (ICA), 54.88 (IJV), 55.52 (VN), 59.64 (skin), and 57.76 (SCM).

To determine the inter-rater reliability of Shore hardness readings, the ICC was calculated using Case 2A ICC(A,1) [19]. Figure 4 presents the results of the ICC study, with the bars representing the ICC value for a given Shore scale and tissue type. The error bars are the 95% CI around the ICC, and the stars indicate statistical significance (*p* < 0.05). As seen in Figure 4, Shore A has a higher inter-rater agreement for tissues compared to the same tissues measured using the Shore OO scale. However, despite the higher agreement, the CIs are wide for all of the ICC values and are also much wider than those predicted during the power calculation.

The overall ICC values can be qualified using different benchmark scales such as Landis and Koch, Fleiss, and Altman. These categorical descriptions of the ICC values can include broad cut-off values and do not take into consideration the number of subjects, raters, categories, or margin of error [21]. For example, all three benchmark scales would qualify the ICC value for the skin using the Shore A scale as substantial (Landis and Koch), excellent (Fleiss), and good (Altman) despite a CI width of 0.53. To mitigate this, ICC qualifications were made by adapting Gwet’s probabilistic approach [21]. Table 5 provides a colour-coded summary of the ICC qualifications after applying the probabilistic approach. As seen in the table, the majority of the ICC values using the Shore A scale are moderate, whilst those using the Shore OO scale are poor.

In summary, with the exception of SCM, hardness measurements of human tissue had better agreement using the Shore A scale compared to measurements on the Shore OO scale. Both scales exhibit wide CI widths, causing increased concern about the reliability of this particular method. For hardness measurements of the SCM, both scales cannot reject the null hypothesis, which indicates no agreement between raters (ICC = 0).

### 3.2. Tissue Thickness and Shore Hardness Values

The correlations between tissue thickness and Shore hardness for Shore A and OO scales are shown in Figure 5. The mean thickness (SD) values were 0.98 mm (0.19) for ICA, 0.33 mm (0.10) for IJV, 1.13 mm (0.26) for VN, 1.58 mm (0.37) for skin, and 5.52 mm (1.17) for SCM.

It should be noted that because the tissue thickness measures for the ICA were not normally distributed, Spearman’s correlation (ρ) was conducted instead of Pearson’s correlation coefficient (*r*); otherwise, all of the associations between thickness and hardness are Pearson’s correlation coefficient.

With the exception of the SCM, all tissues exhibited a negative relationship between thickness and hardness; that is, thicker tissues have lower hardness readings, and thinner tissues have higher hardness readings. Spearman’s correlation demonstrated that the ICA maintained a significant, inverse relationship between tissue thickness and Shore A hardness values (ρ = −0.69, *p* = 0.02).

For Shore OO values, the results were varied. Although the ICA, VN, and skin tissues demonstrated the expected inverse relationships between Shore OO values and tissue thickness, the null hypothesis (*r* = 0) could not be rejected. In fact, with the exception of the IJV, the null hypothesis could not be rejected for any of the correlation coefficients using the Shore OO scale, which indicates that there is no relationship between thickness and hardness. In addition to being significant, the IJV Shore OO results exhibited a positive correlation coefficient. This finding indicates that increases in tissue thickness are correlated with increases in hardness, which is contrary to the Shore A results.

### 3.3. Freehand vs. Stand Measurements

A total of 60 Shore OO hardness readings from the surface of a soft-embalmed liver were measured by a single rater. In total, 30 measurements were taken using a durometer mounted on a stand (mean = 50.80, IQR = 17.40), and 30 measurements were taken freehand (median = 8.55, IQR = 5.50) (Figure 6). A paired Wilcoxon signed-rank test was conducted to evaluate whether there was a significant difference in Shore OO values between measurements taken with the durometer stand and freehand.

The Wilcoxon signed-rank test revealed a significant difference between the two methods of measurement (Z = 4.78, *p* < 0.001), with a rank-based effect size of 0.62. These results are unsurprising given that the median difference between measurements with the stand and freehand was 38.20 Shore OO units. Figure 7 illustrates how different the medians are between the two measurement methods. It is also evident, from Figure 7, that the spread of data is more even than that of the freehand measurement, illustrating the utility of the stand as a method of reducing measurement bias and random error.

## 4. Discussion

This study evaluated (1) the inter-rater reliability of Shore hardness measurements across various tissues, (2) the relationship between tissue thickness and Shore hardness, and (3) differences in measurements obtained using freehand versus mounted durometers. Overall, whilst the inter-rater reliability (ICC values) of clinicians using the Shore A scale was higher than that of the Shore OO scale, the Shore A ICC values were, at best, moderate. Few studies have assessed the inter-rater reliability of Shore hardness among clinicians; however, one relevant study was identified [14]. Similar to this work, it reported mixed reliability and inconsistent correlations between tissue thickness and hardness [14]. Specifically, ICC values ranged from 0.30 to 0.80 for Shore OO measurements taken by two raters on the plantar skin of 20 healthy adults [14].

The findings of the previous study are noteworthy for three reasons. First, the absolute Shore hardness values reported are for fresh, living skin using the Shore OO scale (15–30 [median], 4–41 [min–max]) [14] and are lower than the values reported in this study for embalmed cadaveric tissues (60.17 [median], 52.08–66.46 [min–max]). These discrepancies can likely be attributed to (1) the freehand measurement method used in the previous study [14], (2) differences in skin thickness and region (0.8–1.1 mm in the previous study vs. 1.58 mm in this study), and (3) the use of embalmed tissues in this study.

Second, comparisons of inter-rater reliability between the two studies reveal critical insights. The previous study reported ICC values on plantar skin ranging from 0.30 to 0.80, with confidence interval (CI) values of 0.40 to 0.72 [14]. From these results, four of the six plantar locations were classified as having good or excellent reliability [14]. In contrast, this study reported Shore OO ICC values ranging from 0.00 to 0.40, with CI values of 0.16 to 0.81, across different tissue types in the neck. For the Shore A scale, ICC values ranged from 0.21 to 0.80, with CI values of 0.43 to 0.71. Despite some overlap in ICC ranges between the two studies, this study’s Shore A values were classified as moderate, underscoring how benchmark qualifications can influence the interpretation of results. This raises the question of whether prioritising ICC values near 1.00 with wide CIs is a defensible practise.

Third, the demographics of raters in the previous study were not explicitly reported, which has implications for interpreting ICC results. Rater demographics provide critical information about the generalisability of findings to the intended user population. In this study, the raters represent (1) a potential “floor value” for clinician reliability, as all participants were in the early stages of their medical careers, and (2) the demographic profile of training clinicians likely to use task trainers.

Overall, this study contributes to addressing research gaps in three main ways. First, it emphasises the importance of careful interpretation of inter-rater reliability results when benchmark cut-offs do not account for factors such as the number of subjects, raters, categories, or margin of error. Furthermore, detailed reporting of sample and rater characteristics is essential for ensuring study generalisability. Second, this study highlights the need for implementing or adapting measurement standards, such as ASTM D2240, when evaluating the mechanical properties of tissues. Finally, it provides insights into the impact of tissue thickness and freehand measurement techniques on hardness measurements, which are often used in work utilising Shore hardness as a clinical tool or in the development of tissue mimics for task trainers.

With respect to the results studying hardness and tissue thickness, Shore hardness measurements on the A scale exhibited an expected inverse correlation, consistent with previous studies [22]. In contrast, no consistent relationship between thickness and hardness was observed on the Shore OO scale, suggesting that this scale may be inappropriate for measuring tissue hardness. However, the generalisability of this finding may be limited to tissues from the necks of embalmed cadavers.

Finally, this study underscores the limitations of freehand Shore hardness measurements. Without additional measures to standardise the applied force, freehand methods introduce significant variability, raising concerns about the validity of results. As shown in Figure 7, Shore hardness measurements differed markedly between freehand and mounted durometer methods, both in median values and data distribution. Prior research has established that small variations in applied force can significantly affect Shore hardness readings; for example, a difference of 0.045 N (4.597 g of force) resulted in a 10.27-unit difference in Shore hardness [23]. This highlights both the sensitivity of durometers and the importance of adhering to standardised methods to ensure reliable measurements.

### 4.1. Limitations

This study has several limitations that may have influenced the results and are primarily related to (1) the specimens used, (2) the Shore hardness methodology and equipment, and (3) the raters. The limitations associated with the specimens can be categorised as (1) macro-level limitations, including sample size, donor characteristics, and preservation, and (2) micro-level limitations, such as variations in factors affecting tissue quality (e.g., pathology, collagen content, and senescence).

At the macro level, power calculations indicated that the overall sample size (*n* = 14) was suboptimal, which was evident in the results. For example, using Equation (Equation 1), “reasonable” confidence interval (CI) values for the Shore A ICC results in this work are 0.28 (ICA), 0.30 (IJV), 0.19 (VN), 0.32 (skin), and 0.08 (SCM). However, the actual CI values for Shore A measurements ranged from 0.48 to 0.73, as shown in Figure 4. For the Shore OO scale, the mismatch was even greater. To achieve reasonable CI values for the Shore A scale, sample sizes of 16–70 would be required. For the Shore OO scale, excluding the SCM, sample sizes would need to range from 30 to 170. Notably, no reasonable CI value could be calculated for the SCM on the Shore OO scale given the study results, highlighting that the study was likely underpowered.

In addition to the small sample size, donor variability likely contributed to the wide CIs. Human tissues exhibit considerable variability in mechanical properties due to structural protein content, age, sex, and underlying diseases. For instance, arterial calcification is a normal part of ageing and is exacerbated by conditions such as atherosclerosis and hyperlipidaemia [24]. This introduces two major implications in our study: (1) variability in hardness may result from unaccounted disease states, and (2) the findings may not generalise to populations with different demographic profiles, particularly younger or healthier individuals. It should be noted that stratifying or controlling for vessel calcification was not feasible due to unavailable donor medical histories, lack of concurrent histopathological analyses, and the inevitable presence of calcification in our donor population,

Another factor impacting generalisability is that all tissues in this study were preserved. Previous work reported a mean Shore OO value of 41 for arterial tissues [25], compared to this study’s mean of 54.88. This suggests that Shore OO may be more reliable for fresh or living tissues, whilst Shore A might be more suitable for preserved tissues. Future studies should explore the reliability of these scales for unpreserved tissues.

A further limitation was the relationship between tissue dimensions and durometer readings. During indentation, the durometer’s foot induces localised plastic deformation, which affects not only the material beneath the indentation but also the surrounding material because of the increases in strain-induced dislocation density [26]. Testing standards account for such effects by prescribing minimum distances from edges of the tested material and material thickness. However, most of the tissue dimensions in this study did not meet these standards, potentially affecting both the precision and validity of the results. Future research should investigate how tissue dimensions impact hardness measurements and determine whether these effects significantly alter the required precision for mimic development.

The use of a digital durometer was another limitation. A post hoc investigation revealed potential digital drift in the two durometers used. However, it was too difficult to distinguish between the effects of drift and rater variability. For example, Shore A reading fluctuated and generally increased over time, but the raters did not take measurements of the same tissues on the same day to ensure blinding. Future studies should consider (1) whether analogue durometers provide better reliability than digital durometers, and (2) controlling for digital drift as much as possible.

Finally, there were limitations in the number and selection of raters. Only three raters were used for ICC calculations. Such a small number of raters may not fully capture the variability across different users, which may not fully capture the variability across different users and thus impacts the generalisability of inter-rater reliability results. Furthermore, another limitation was in the ICC calculation. A two-way random effects layout was employed, but it could be argued that the raters were not truly random representatives of the overall population collecting Shore hardness measurements. Future studies may want to consider the impacts of employing random effects versus mixed-effect layouts in inter-rater reliability studies using Shore hardness.

### 4.2. Future Considerations and Research

The limitations highlighted in the previous section underscore areas for improvement, particularly related to (1) sample size and characteristics, (2) accuracy and precision of digital durometers, and (3) the number and selection of raters.

Previous studies have investigated the use of Shore hardness durometers as a clinical tool due to their simplicity, non-destructive nature, and ease of use for freehand measurements [9,11,13,14]. However, this study serves as a cautionary tale, emphasising the importance of carefully considering the available standards, Shore scales, and the reliability of the employed methods. Future studies where hardness is a critical measure should investigate the effects of rater training, expertise, and environmental factors (e.g., clinical vs. research settings) on inter-rater reliability. Additionally, refined methods involving multiple raters and applied force measuring tools could provide more insight into the accuracy and precision of durometers as clinical tools when using a freehand approach.

Whilst some groundwork has been laid to relate different biomechanical properties to Shore hardness [23,27,28], more work is needed. Future directions include exploring the relationships between Shore hardness and other mechanical properties, such as viscoelasticity or creep, through different types of loading (e.g., compression or shear testing). Including a broader variety of tissue types (e.g., cartilage, tendons, neurovascular tissue) and employing different Shore scales could provide deeper insights into the generalisability of Shore hardness. Furthermore, given that both mechanical properties and geometry of tissues are heterogeneous, standardisation efforts in collaboration with biomechanical and material science organisations would also help to ensure reproducible and reliable results in future studies.

The relationship between mechanical properties and tactile perception is another area that warrants further exploration. Establishing a deeper understanding of this relationship could enhance clinical assessment tools and training methodologies. Future studies using Shore hardness as a basis for mimicking tissue properties should incorporate functional testing that replicates clinical applications, such as needle puncture, suturing, incision force, and palpation. Such work could also expand to (1) determine whether Shore hardness is a reliable measure of haptic fidelity and (2) evaluate training outcomes (e.g., skill acquisition and error reduction) in procedural task trainers specifically designed to mimic the Shore hardness of tissues.

## 5. Conclusions

In conclusion, this study highlights the need for caution when using Shore hardness to select tissue mimics. Whilst it is an accessible and convenient method, the findings of this work demonstrate that Shore hardness measurements are influenced by the Shore scale, tissue type, and tissue thickness. Overall, clinician reliability using the Shore A scale was found to be moderate for vascular tissues and skin but poor for nervous and muscle tissues. For the Shore OO scale, reliability was consistently poor across all tissue types. These results, however, may not be generalisable due to the use of tissues from embalmed cadavers. Further investigations are needed to elucidate whether these findings are true for fresh tissues.

Despite this limitation, the study underscores the importance of standardised methods, such as ASTM D2240, and advocates for the development of standards developed by a multidisciplinary group to address the challenges of measuring the Shore hardness of biological tissues. This work provides critical insights for clinicians considering the use of Shore hardness durometers in clinical settings or for the development of training tools. Future research should focus on (1) correlating functional tests (e.g., suturing, palpation) with Shore hardness, (2) evaluating Shore hardness as a measure of haptic fidelity, and (3) assessing the training outcomes of devices that utilise Shore hardness to replicate human tissues.

## Figures and Tables

**Figure 1 bioengineering-12-00041-f001:**
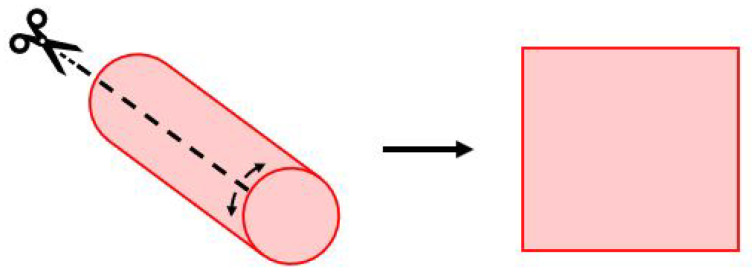
Longitudinal opening of the ICA and IJV for hardness testing.

**Figure 2 bioengineering-12-00041-f002:**
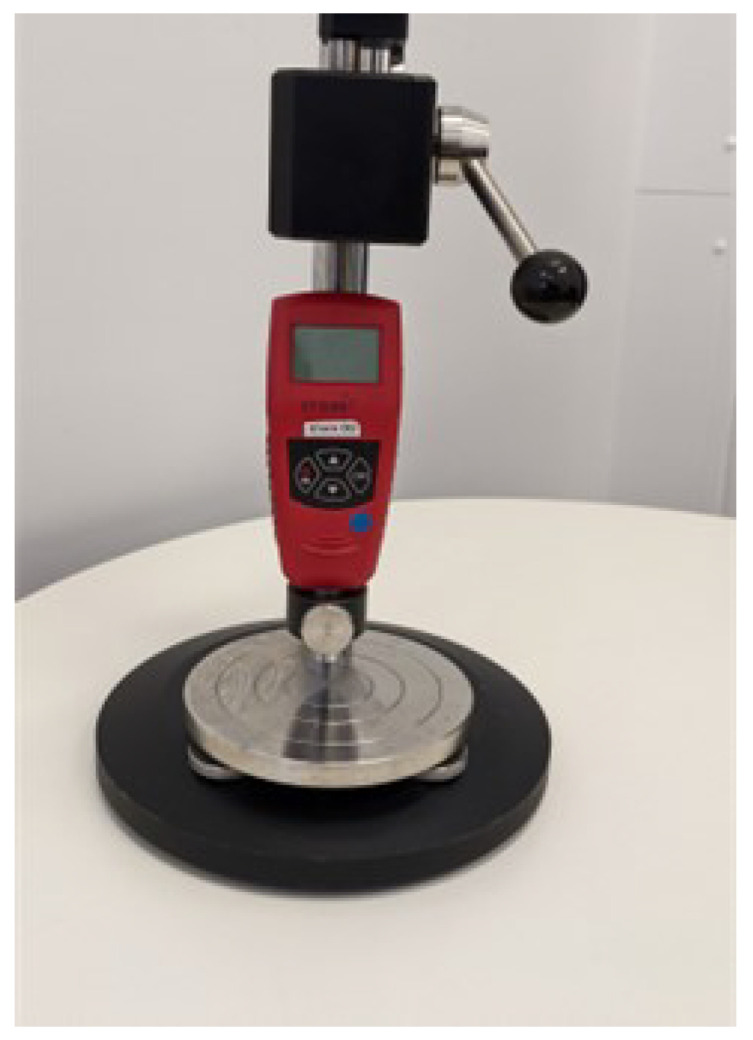
Digital durometer and stage with weight used to collect Shore hardness specimens.

**Figure 3 bioengineering-12-00041-f003:**
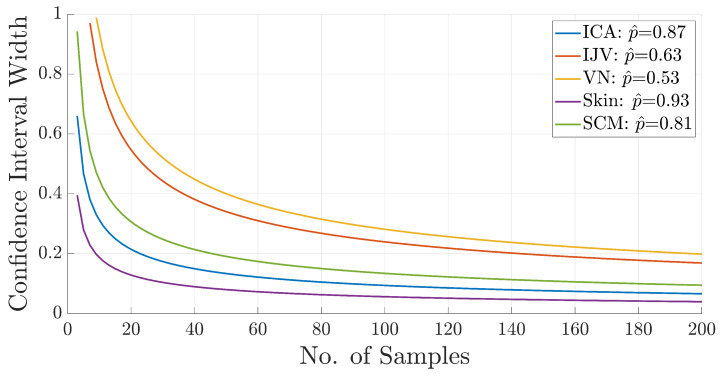
Plot of the expected confidence interval width vs. number of samples. Calculations were based on the absolute agreement coefficient ($\hat{p}$) for each tissue type.

**Figure 4 bioengineering-12-00041-f004:**
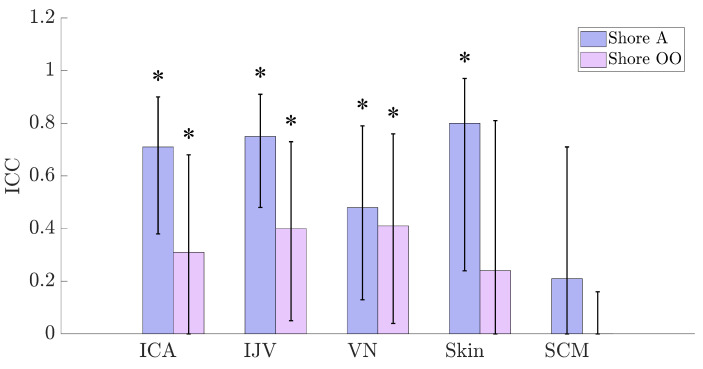
ICC values, CI, and significance for the measured hardness of different human tissues using the Shore A and OO scales. The asterisks (*) indicate significant ICC values (*p* < 0.05).

**Figure 5 bioengineering-12-00041-f005:**
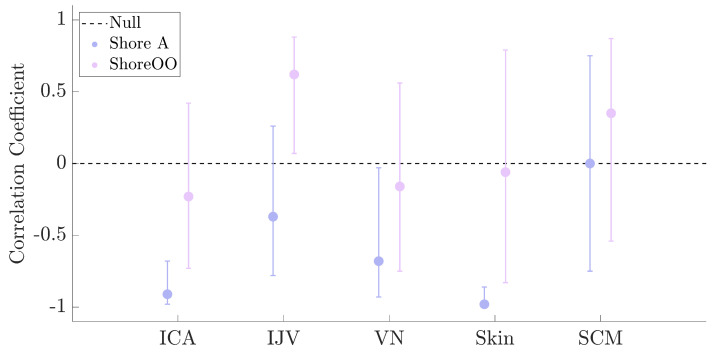
Correlation coefficients for the association between tissue thickness (mm) and Shore scale are represented by the filled circles. The bars around the circles are the 95% CI of the correlation coefficients.

**Figure 6 bioengineering-12-00041-f006:**
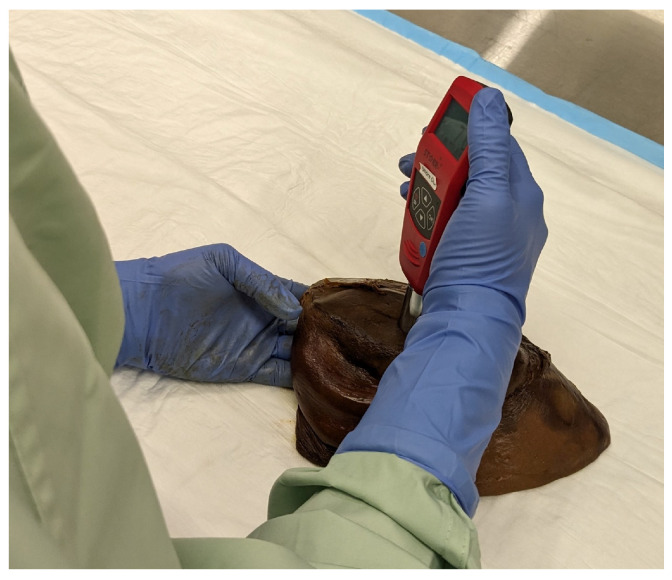
Rater taking freehand Shore OO measure from the surface of a liver.

**Figure 7 bioengineering-12-00041-f007:**
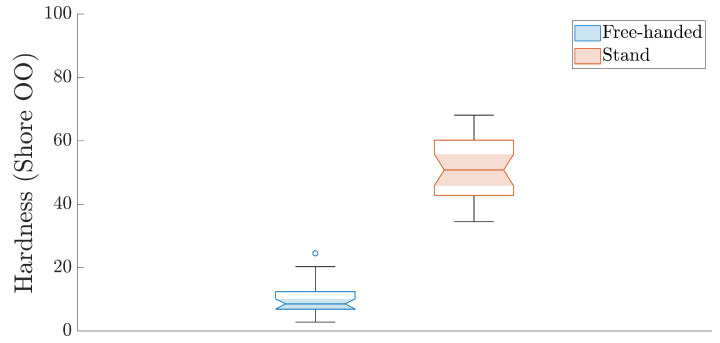
Notched boxplot showing the data distribution for Shore OO measurements on the surface of a liver.

**Table 1 bioengineering-12-00041-t001:** Shore hardness values for biological tissues in the literature.

Tissue	Shore A	Shore O	Shore OO
Heart	7.5–18.0 [10]	43.0 [8]	68.0 [8]
Liver		18.0 [8]	15.1–47.0 [8,11]
Spleen		7.0 [8]	16–40.5 [8,9]
Kidney		32.0 [8]	58.0 [8]
Brain		5.0 [8]	38.0 [8]
Pancreas		13.0 [8]	27.0–65.8 [8,9]
Skin	13.3–28.0 [12]	28.0–71.0 [13]	

**Table 2 bioengineering-12-00041-t002:** Characteristics of cadavers from which tissue samples were obtained. For mean values, the standard deviation is presented in parentheses.

Specimen Characteristics	
No. of specimens	14
% Female	36
Mean age in years	86.7 (6.2)
Mean number of days from death to embalming	2.6 (0.9)

**Table 3 bioengineering-12-00041-t003:** Characteristics of the raters collecting Shore hardness measurements.

Rater	Education ^a^	Relevant Work Experience ^b^	Age
1	PhD candidate ^c,^* M.Sc. MD2 * BSE	Staff scientist TA RA	32
2	M.Sc. MHSPAS MD1 * B.Sc.	Physician assistant RA TA	33
3	M.Sc. MD2 * B.Sc.	Physiotherapist assistant RA TA	30

^a^: Bachelor of Science (B.Sc.), Bachelor of Science in Engineering (BSE), medical student year 1, 2 (MD1, 2), Master of Science (M.Sc.), Master of Health Science Physician Assistant Studies (MHSPAS); ^b^: research assistant (RA), teaching assistant/tutor (TA); ^c^: * degree ongoing during data collection.

**Table 4 bioengineering-12-00041-t004:** Estimations of sample sizes that will provide a “reasonably” narrow CI width for each tissue type.

	$\hat{p}$	${\mathrm{CI}}_{\mathrm{Narrow}}$	*n*
ICA	0.87	0.35	10
IJV	0.63	0.25	100
VN	0.53	0.21	>200
SCM	0.81	0.37	14
Skin	0.93	0.32	5

**Table 5 bioengineering-12-00041-t005:** Shore A and OO hardness values for each tissue type with upper and lower 95% CIs in parentheses, respectively. Each tissue type contains adjusted benchmarks, which are shown for Landis and Koch (LK), Fleiss (Fl), and Altman (Al) for each scale. The benchmarks are colour-coded for ease of visualisation and the qualifications are as follows: moderate (M), intermediate/good (IG), slight (S), and poor (P).

Tissue	Scale and Benchmarks					
	**A**			**OO**		
	**LK**	**Fl**	**Al**	**LK**	**Fl**	**Al**
ICA	0.71 (0.38, 0.90)			0.31 (0, 0.68)		
	M	IG	M	P	P	P
IJV	0.75 (0.48, 0.91)			0.40 (0.05, 0.73)		
	M	IG	M	P	P	P
VN	0.48 (0.13, 0.79)			0.41 (0.04, 0.76)		
	S	P	P	P	P	P
Skin	0.80 (0.24, 0.97)			0.24 (0.00, 0.81)		
	M	IG	M	P	P	P
SCM	0.21 (0.00, 0.71)			0.00 (0.00, 0.16)		
	P	P	P	P	P	P

## Data Availability

The raw data are available for the following datasets: reliability calculations (https://osf.io/fmh25), freehand vs. stand (https://osf.io/83chv), and tissue dimensions (https://osf.io/c89qu).

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
