# Peer review of "Investigating the Reliability of Shore Hardness in the Design of Procedural Task Trainers"

_bioengineering, 2025, doi:10.3390/bioengineering12010041_

Round 1
Reviewer 1 Report
Comments and Suggestions for Authors
The presented article focuses on the crucial topic of haptic fidelity in procedural task trainers, which is critical for improving medical training and patient safety. The authors' investigation of the dependability of Shore hardness measurements for human tissues provides useful insights into the selection of biomimetic materials, revealing both obstacles and potential solutions in medical simulation design. The findings could have a substantial impact on the development of more effective training tools for healthcare practitioners. However, the following comments need to be considered before accepting for publication:
Abstract
- The mention of "poor to moderate" reliability could be quantified with specific ICC values to provide a clearer understanding of the study's findings.
Introduction
The introduction may benefit from a more explicit articulation of the research gap. While the authors mention the "limited research on the reliability of human tissue measurements using Shore scales," they should elaborate on existing studies to highlight what specific gaps this research addresses.
Materials and Methods
The methods section lacks detail in the statistical analysis approach. The authors should provide more information on how the sample size was determined, including any power calculations. For example, while the authors mention a "realistic sample size of 14 cadavers," it would be beneficial to explain how this number was derived and its implications for the study’s power.
Results
- The results section could be improved by integrating visual aids to summarize key findings. For instance, including a table that summarizes the ICC values alongside their confidence intervals for each tissue type would allow for easier comparison and interpretation.
- The authors should discuss the implications of the findings in relation to existing literature, particularly how the inter-rater reliability compares to those reported in previous studies.
Discussion
- The discussion effectively interprets the results, but it could benefit from a more critical evaluation of the limitations. While the authors mention "inconsistency in tissue thickness," they should provide specific examples of how this inconsistency may have affected the results.
- Discussing future research directions would provide a more comprehensive view of the study's impact.
Conclusion
The conclusion should emphasize the practical implications for the design of procedural task trainers. The authors should explicitly state how their findings could influence future research or clinical practice in medical training.
Author Response
Thank you for your thoughtful and thorough review of our manuscript. We sincerely appreciate your valuable insights and constructive feedback, which have significantly contributed to enhancing the quality and scientific rigor of our work. Your comments are addressed below.
Comment1: Abstract: The mention of "poor to moderate" reliability could be quantified with specific ICC values to provide a clearer understanding of the study's findings.
Response1: The abstract has been thoroughly revised (lines 1-23) to include specific ICC values, correlation coefficients, and rank testing results. Additionally, the conclusion sentence has been updated to align more closely with the revised discussion and conclusions presented in the paper.
Comment 2: Intro: The introduction may benefit from a more explicit articulation of the research gap. While the authors mention the "limited research on the reliability of human tissue measurements using Shore scales," they should elaborate on existing studies to highlight what specific gaps this research addresses.
Response 2: The last two paragraphs of the introduction (lines 87-103) have been revised to explicitly articulate the research gap and segue into the aims and hypothesis of this paper. Specifically, we added: "As use of physical simulations grows, particularly in the context of 3D printing and do-it-yourself approach, the demand for standardised tissue values becomes increasingly important. Whilst Shore hardness is an appealing mechanical metric for material selection in clinical training tools, some studies have briefly explored its ability to inform clinical decision making [9,11,13,14]. Therefore, it is also imperative to establish whether measuring Shore hardness is a reliable method for use among a clinician population, as there are significant research gaps in this area."
Comment 3: Methods: The methods section lacks detail in the statistical analysis approach. The authors should provide more information on how the sample size was determined, including any power calculations. For example, while the authors mention a "realistic sample size of 14 cadavers," it would be beneficial to explain how this number was derived and its implications for the study’s power.
Response 3: The subsection on power calculation (lines 160-186) have been revised to:
- Detail how the power calculations were estimated.
- Provide a thorough discussion of the curves in Figure 3.
- Include a new table (Table 4) summarising the "reasonable" confidence interval widths and corresponding sample sizes.
- Explain how the "realistic sample size" of 14 cadavers was determined, balancing power and resource limitations.
- Additionally, we expanded the discussion section to include the limitations of the selected sample size (lines 340–349).
Comment 4: Results: The results section could be improved by integrating visual aids to summarize key findings. For instance, including a table that summarizes the ICC values alongside their confidence intervals for each tissue type would allow for easier comparison and interpretation.
Response 4: Data visualisation has been extensively revised, as outlined below:
- Figure 4 (revised figure): The original boxplots have been replaced with a new figure (pg. 8) hat highlights ICC differences between the two Shore scales. Error bars illustrate confidence intervals. Mean hardness values for each tissue type are now included in the narrative (lines 212–215).
- Tables 4 and 5 (original tables): These have been combined into a single table (Table 5, pg. 9), presenting ICC values, 95% CI, and color-coded benchmarks. This approach contextualises numerical values, emphasising the importance of probabilistic benchmarking (lines 221–232).
- Figure 5 (revised figure): Replacing Table 6 in the original manuscript, this new figure plots correlation coefficients with 95% CI, with null values included for significance visualisation. Mean tissue thickness values are now provided in the narrative (lines 240–242).
Comment 5: Results: The authors should discuss the implications of the findings in relation to existing literature, particularly how the inter-rater reliability compares to those reported in previous studies.
Response 5: The discussion now includes a comparison to the limited existing literature on inter-rater reliability of Shore hardness measurements in human tissues (lines 279–307). We also summarise this study's contributions to ICC interpretation, generalisability, and standardisation (lines 308–314).
Comment 6: Discussion: The discussion effectively interprets the results, but it could benefit from a more critical evaluation of the limitations. While the authors mention "inconsistency in tissue thickness," they should provide specific examples of how this inconsistency may have affected the results.
Response 6:
We have significantly expanded the limitations discussion (lines 334–390), broadly categorizing them into:
- Specimens:
- Macro-level limitations (e.g., underpowered sample size, lines 340–348).
- Macro- and micro-level limitations (e.g., variability in tissue quality due to age or pathology, lines 350–359).
- Shore hardness/durometer methodology:
- Challenges related to non-standardised tissue dimensions (lines 365–374).
- Issues with digital drift affecting measurement consistency (lines 375–381).
- Raters:
- Limited number and representativeness of raters, as well as ICC selection considerations (lines 382–390).
Comment 7: Discussion: Discussing future research directions would provide a more comprehensive view of the study's impact.
Response 7:
A new subsection on future considerations and research directions (lines 392–422) has been added, emphasising:
- Investigating factors affecting inter-rater reliability, such as rater training and environmental conditions (lines 395–403).
- Exploring Shore hardness as a metric for tissue mechanical properties across different regions and tissue types (lines 404–412).
- Developing task trainers incorporating Shore hardness and functional testing to evaluate clinical tasks and training outcomes (lines 414–422).
Comment 8: Conclusion: The conclusion should emphasize the practical implications for the design of procedural task trainers. The authors should explicitly state how their findings could influence future research or clinical practice in medical training.
Response 8: The conclusion (lines 424–441) has been rewritten to:
- Summarise the study's findings, highlighting the influence of Shore scale, tissue type, and tissue thickness on reliability (lines 425–432).
- Advocate for the development of Shore hardness standards for biological tissues (lines 433–436).
- Outline future research directions, including correlating Shore hardness with functional tests and evaluating training outcomes (lines 436–441) in the context of task-trainer development.
Reviewer 2 Report
Comments and Suggestions for Authors
The abstract is well-structured and summarizes the study's objectives, methods, and conclusions.
The importance of Shore hardness measurements in biomaterials and their associated challenges are highlighted.
The need for biomimetic materials and the application of Shore hardness measurements are clearly presented.
The context of the use of medical simulators is well explained.
The use of the ASTM D2240 standard and the explanations about sample thickness are well-founded.
The results are well organized, with diagrams and tables that facilitate interpretation, but the wide ICC for some measures suggests methodological limitations that are not sufficiently discussed.
The statistical conclusions are presented clearly, highlighting the differences between the measurement methods.
Overall, the paper provides a solid foundation for further research and is relevant for the development of more realistic and effective medical simulators. Implementation of the recommendations in the article has the potential to improve professional medical training significantly.
Author Response
Thank you for your thoughtful and thorough review of our manuscript. We sincerely appreciate your valuable insights and constructive feedback, which have significantly contributed to enhancing the quality and scientific rigor of our work. Your comments are addressed below.
Comment 1: Results: The results are well organized, with diagrams and tables that facilitate interpretation, but the wide ICC for some measures suggests methodological limitations that are not sufficiently discussed.
Response 1: We have significantly expanded the discussion section to include a subsection for limitations (lines 334–390), broadly categorising them into:
- Specimens:
- Macro-level limitations (e.g., underpowered sample size, lines 340–348).
- Macro- and micro-level limitations (e.g., variability in tissue quality due to age or pathology, lines 350–359).
- Shore hardness/durometer methodology:
- Challenges related to non-standardised tissue dimensions (lines 365–374).
- Issues with digital drift affecting measurement consistency (lines 375–381).
- Raters:
- Limited number and representativeness of raters, as well as ICC selection considerations (lines 382–390).
Round 2
Reviewer 1 Report
Comments and Suggestions for Authors
The authors have addressed all the reviewer's concerns and the article is ready for publication.